

# Network neighborhood operates as a drug repositioning method for cancer treatment

Ali Cüvitoğlu[1] and Zerrin Isik[2]

[1] The Graduate School of Natural and Applied Sciences, Dokuz Eylül University, Izmir, Turkiye
[2] Computer Engineering Department, Engineering Faculty, Dokuz Eylül University, Izmir, Turkiye

## ABSTRACT

Computational drug repositioning approaches are important, as they cost less compared to the traditional drug development processes. This study proposes a novel network-based drug repositioning approach, which computes similarities between disease-causing genes and drug-affected genes in a network topology to suggest candidate drugs with highest similarity scores. This new method aims to identify better treatment options by integrating systems biology approaches. It uses a protein-protein interaction network that is the main topology to compute a similarity score between candidate drugs and disease-causing genes. The disease-causing genes were mapped on this network structure. Transcriptome profiles of drug candidates were taken from the LINCS project and mapped individually on the network structure. The similarity of these two networks was calculated by different network neighborhood metrics, including Adamic-Adar, PageRank and neighborhood scoring. The proposed approach identifies the best candidates by choosing the drugs with significant similarity scores. The method was experimented on melanoma, colorectal, and prostate cancers. Several candidate drugs were predicted by applying AUC values of 0.6 or higher. Some of the predictions were approved by clinical phase trials or other *in-vivo* studies found in literature. The proposed drug repositioning approach would suggest better treatment options with integration of functional information between genes and transcriptome level effects of drug perturbations and diseases.

## INTRODUCTION

Drug repositioning (DR) aims to find a new use for existing approved drugs in the treatment of different diseases. In recent decades, DR has been used in the search for novel cancer treatments that have high mortality rates, and is a popular alternative to the development of entirely new drugs. Developing a new drug traditionally is a costly process and is quite time consuming, whereas DR gives higher success rates relatively in shorter times.

There are two general DR approaches: experimental and computational. Duloxetine, sildenafil, and thalidomide are some of the first experimental examples of DR that have achieved clinical success (*Thor & Katofiasc, 1995*; *Renaud & Xuereb, 2002*; *Stephens & Brynner, 2009*; *Tansey, 2001*). Experimental approaches may be very successful for the

Corresponding author
Zerrin Isik, zerrin@cs.deu.edu.tr

repositioning of a drug, however, high numbers of FDA-approved drugs and potentital disease states make it impossible to test all drug-disease combinations with experimental methods. For this reason, finding the best possible estimates using computational methods is gaining serious attention. Phenoxybenzamine, sulconasone (topical antifungal), and vinburnin (vascular expander) are some of the successful treatments identified by computational methods (*Chang et al., 2010*; *Iskar et al., 2013*). The use of computational drug repositioning (CDR) methods has increase recently, current approaches apply machine learning (ML) and biological network integration. PREDICT is an ML model used to find new associations between drugs and diseases (*Gottlieb et al., 2011*). Firstly, they collected the data from OMIM, DrugBank, DailyMed, and Drugs.com. After the construction of drug–drug and disease–disease similarities, these similarities wre exploited to construct classification features and the subsequent learning of a classification rule. A 0.90 AUC was obtained in their experiments with a 10-fold cross validation and new drugs were proposed for many diseases listed in OMIM. Other DR approaches have applied ML models (*Aliper et al., 2016*; *Chyr, Gong & Zhou, 2022*; *Yang et al., 2022*). Although machine learning is one of the most remarkable methods of recent times, some shortcomings still exist. For example, an unbalanced data set or small number of samples may lead over-fitting of machine-learning methods. Different computational drug repositioning methods have been applied during the COVID-19 pandemic (*Yang et al., 2022*). A machine learning DR study integrated knowledge graphs, literature, transcriptome data, and repurposed the CVL218 compound for the treatment of COVID-19 by providing *in-vitro* evidence (*Ge et al., 2021*). Another machine learning approach evaluates FDA-approved broad-spectrum antiviral drugs by computing network regulated effects on the COVID-19 disease module (*de Siqueira Santos et al., 2022*).

Recently, network-based methods have gained more attention (*Lotfi Shahreza et al., 2017*). In these approaches, interactions are used to present a physical relationship between two proteins or a functional similarity between genes within a biological network, which may represent more than one type of relationship at the same time (*Zou et al., 2013*; *Rider, Chawla & Emrich, 2013*). Network structures represent different biological interactions that include gene regulatory networks, metabolic networks, protein-protein interaction (PPI) networks, drug-target/drug-drug/drug-disease/side-effect relationships or disease-disease relationships (*Lotfi Shahreza et al., 2017*). Gene expression measurements in the transcriptome level for drug-treated cells can provide insights about cell's dynamic response to the treatment and molecular mechanisms triggered by drugs (*Dai & Zhao, 2015*). Differential expression profiles of genes vary between disease and control samples. There are many studies using differential gene expression profiles as fundamental input to prioritize potential drug targets (*Chang, Shoemaker & Wang, 2011*; *Yeh, Yeh & Soo, 2012*; *Isik et al., 2015*; *Chen et al., 2016b*). The Functional Module Connectivity Map (FMCM) has been designed and used as a DR method in colorectal adenocarcinoma (*Chung et al., 2014*). Besides this, there are many studies that have achieved success using a network-based approach. Another study known as MNBDR used protein-protein interactions and gene expression profiles to predict drug candidates for 19 cancer datasets (*Chen & Zhou, 2021*). Another study applied three signatures (chemical structure, drug-target association, and

gene expression of drug treatment) (*Lee, Kang & Kim, 2016*). Drugs were repositioned based on the signature similarities of drugs and diseases. The classifiers with structure or target signatures achieved only 0.62 AUC, however the expression signature reached up to 0.79 AUC for various cancers. A DR study computed a correlation score between functional networks of diseases and drug perturbations (*Chen et al., 2016a*). They reported repositioned drugs for prostate (0.51−0.69 AUC) and breast (0.51−0.75 AUC) cancers by using the LINCS drug expression profiles. The NEDNBI model built a gene-disease-drug network, then applied diffusion process in this network to predict new interactions between diseases and drugs (*Qin et al., 2022*). The model was evaluated by a 10-fold cross validation on several diseases obtained from repoDB and DisGeNet databases. It also proposed 20 potential treatments for COVID-19. There are also hybrid studies using both machine-learning and network-based methods (*Bahi & Batouche, 2018*; *Zeng et al., 2019*; *Luo et al., 2019*; *Galan-Vasquez & Perez-Rueda, 2021*; *Meng et al., 2022*; *Yan et al., 2022*). Network-based methods are effective for finding new biological modules, however there is no gold standard to test associations among biological modules.

This study proposes a novel network-based repositioning approach using different data sources such as functional interaction networks, drug-treated transcriptome profiles, and disease-causing genes. A functional interaction data between proteins is used as the main network structure. Network neighborhood metrics are adapted to compute a similarity score between the disease-causing network and the drug-affected network. Neighborhood metrics are adapted to utilize differential gene expressions that are obtained from transcriptome data of drug-treated cells and related patient cohorts. Experimental results and their computational validations are provided for colorectal cancer, prostate cancer, and melanoma. The fundamental differences of the proposed DR method include the integration of gene expression changes into network neighborhood metrics and the representation of drug-specific expression perturbations as functional network modules.

## METHOD

The proposed DR model is based on several network structures that consider the assembly of disease-causing genes or drug-affected proteins. A functional interaction network (FIN) was obtained from the literature. Gene expression data for drug-treated cells were downloaded from the LINCS project. Differentially expressed genes (DEG) were obtained for each drug-treated cancer cell line. Finally, the DEG set of each drug sample was mapped on the FIN by using a direct neighbor mapping and drug specific functional interaction networks (drug-affected protein network; DAPN) were obtained (Figs. 1A and 1C). A different kind of disease-causing gene (DG) was retrieved from the TCGA project (*National Cancer Institute, 2023*) using the TCGABiolinks package (*Mounir et al., 2019*). After applying statistical analysis to calculate differential mRNA expression for each cohort, DG was mapped on the FIN using the direct neighbor mapping, this network was named the disease genes network (DGN) (Figs. 1B and 1D). The similarity between the DAPN and DGN modules is calculated according to a combined score based on topological closeness and biological function similarities (Fig. 1E). For this calculation, three network metrics
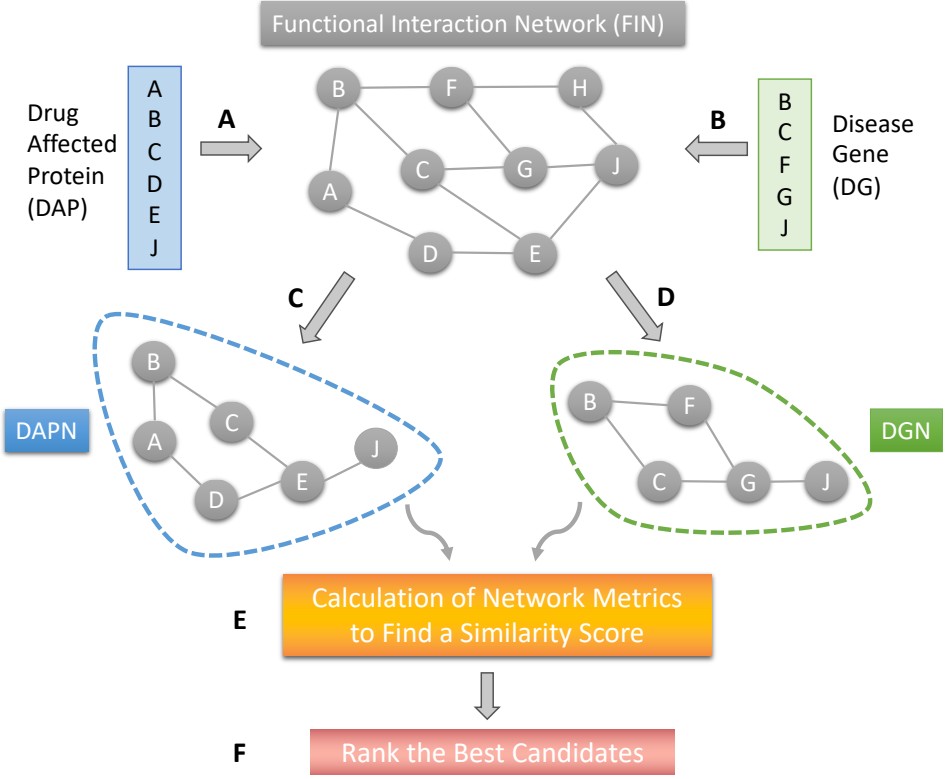

**Figure 1** **The workflow of the proposed model.** (A) Mapping of drug affected genes on FIN. (B) Mapping of cancer causing genes on FIN. (C) Each drug-affected protein network (DAPN) module is extracted by considering the direct neighbors of drug-affected proteins. (D) A DGN module is created by considering direct neighbors of disease-causing genes. (E) Network metrics run on DAPN and DGN to find a similarity score by combining AUC scores. (F) The best candidates are listed by ranking the drugs with highest AUC scores.

were applied: Adamic-Adar Coefficient, PageRank, and Neighborhood Scoring. Finally, a DAPN-DGN similarity score was computed for each metric separately to get an AUC value of the similarity between the DGN and DAPN modules (Fig. 1E). The similarity scores (in terms of AUC) were ranked from highest to lowest (Fig. 1F). As a computational validation, the top-predictions were compared with the known FDA-approved treatments and clinical studies found for the related cancer.

## Functional interaction network

The functional interaction network (FIN) contains the functional interaction data of proteins and was obtained from the literature (*Linghu et al., 2009*). The FIN consists of 20,790 unique proteins and 21,952,150 interactions. The biological similarity between the two proteins was represented as an interaction weight. In order to understand the interaction between the proteins, the weight distribution was examined. Figure S1 shows the distribution of weights and expresses biological similarities between proteins, with an accumulation between 0–0.1. Due to the representing very low biological similarities up to the 0.1 value, these interactions were excluded from FIN. A pruned FIN was obtained by

excluding links with less than 0.1 weight value, which contains 15,002 proteins and 334,225 interactions.

## Drug treated samples

Transcriptome data for drug-treated cancer cell lines are available in the LINCS project (*Subramanian et al., 2017*). This project covers drug-treated experiments for several dosages and cancer cell lines. The Genomics of Drug Sensitivity in Cancer (GDSC) (*Yang et al., 2012*) and Chembl databases (*Mendez et al., 2019*) were used to determine the optimal dosage for each drug in related cancer types. If an optimal IC50 dosage of a drug was obtained from these databases, this specified dosage of the drug was used in the proposed model. Using the selected IC50 doses of drugs, drug-treated cell line data was downloaded from the LINCS project. The total number of differentially expressed genes for each drug sample was experimentally decided by applying several z-score thresholds such as |1|, |1.5| and |2|. Filtering the z-score > |1| represented a more significant number of DEGs for each drug and drug-affected proteins (DAP) were obtained by filtering the DEG with the z-score > |1|. The resulting protein list was mapped to the FIN using a direct neighborhood method. Finally, a drug-affected protein network (DAPN) was individually constructed for each drug. Details of this network mapping procedure was explained in our previous work (*Cuvitoglu & Isik, 2020*). DAPNs was obtained that cover 260, 243, and 220 different drugs for colorectal cancer (HT-29), prostate cancer (PC3), and melanoma (A375) cell lines, respectively.

## Disease-causing genes

The RNA-sequencing data of colorectal cancer, prostate cancer, and melanoma was used to identify disease-causing genes for each cancer. For this purpose, the COAD, PRAD, and SKCM cohorts were downloaded from the TCGA project (*National Cancer Institute, 2023*) using the TCGABiolinks (*Mounir et al., 2019*) package. Primary solid tumor (TP) and solid tissue normal (NT) samples were extracted for each cancer type. The total number of TP samples were 285, 497, and 103; the number of NT samples were 41, 52, and 1 for COAD, PRAD, and SKCM cohorts, respectively. A statistical analysis pipeline was applied to calculate mRNA expression changes for each cohort. The FDR threshold value of 0.01 and the absolute fold-change (FC) threshold value of 2.0 were used to identify differentially expressed genes (DEG). The total number of DEG are 1,198, 319, 133 for colorectal cancer, prostate cancer, and melanoma, respectively. These procedures resulting in the construction of an individual disease gene network (DGN) for each cancer. The DGNs were covering 813, 157, and 73 proteins for colorectal cancer, prostate cancer, and melanoma, respectively.

## Network metrics

Three different network metrics were applied to calculate a similarity score based on topological closeness and biological function similarity between the network of disease genes (DGN) and the network of drug-affected proteins (DAPN). Disease genes and drug-affected proteins may not show a high overlap in terms of exact gene matching, since gene expression changes might occur in close neighborhood or downstream partners.
Therefore computing a closeness score between these genes in a network topology may model biochemical responses in a cellular environment more realistically. The Adamic-Adar coefficient, PageRank, and neighborhood scoring are known metrics applied to find central nodes and to suggest new potential interactions between nodes or to explore vicinity of nodes in a network. These metrics were adapted to use the interaction weight (as a biological function) and to integrate z-scores of genes (differential expression) into the similarity score calculation.

### Adamic-Adar

This formula is used to calculate the degrees of common neighbors of two genes (*Adamic & Adar, 2003*).

$$S_{xy} = \sum_{z \in \lambda x \cap \lambda y} 1/\log k_z, \tag{1}$$

where $x$ and $y$ are different genes, $S_{xy}$ represents the similarity between the gene $x$ and $y$. $z$ is a neighbor node of both $x$ and $y$. $k_z$ is the degree of the node $z$. To compute this metric, the '*similarity*' function in the '*igraph*' package was applied. In this function, the method parameter was set to '*invlogweighted*' to compute the Adamic-Adar coefficient. At this stage, only the similarity between two genes is obtained, therefore a matrix should be formed when it runs for all gene pairs. The sum of all of the values of $S_{xy}$ was used to generate a similarity score for a gene $x$. Equation (2) shows that the similarity score ($S_x$) of the gene $x$ is expressed by the sum of its similarities with all other $y$ genes in the network.

$$S_x = \sum_{y \in N} S_{xy}, \tag{2}$$

where $N$ represents all genes in the network. $S_{xy}$ represents the similarity between $x$ and $y$ genes; $y$ represents other genes in the network. After obtaining the Adamic-Adar coefficient ($\forall S_x$) on a gene basis, new formulations ($AA_1$ and $AA_2$) were created to integrate differential expression of genes into the similarity calculation (Eqs. (3) and 4).

$$AA_1(x) = \begin{cases} S_x * 0.5^{\|z(drug,x)+z(disease,x)\|}, & \text{if } x \in (DGN \text{ \& } DAPN) \\ S_x * 0.5^{\|z(drug,x)\|}, & \text{if } x \in (DAPN) \\ S_x * 0.5^{\|z(disease,x)\|}, & \text{if } x \in (DGN) \end{cases} \tag{3}$$

$$AA_2(x) = \begin{cases} S_x / e^{\|z(drug,x)+z(disease,x)\|}, & \text{if } x \in (DGN \text{ \& } DAPN) \\ S_x / e^{\|z(drug,x)\|}, & \text{if } x \in (DAPN) \\ S_x / e^{\|z(disease,x)\|}, & \text{if } x \in (DGN) \end{cases} \tag{4}$$

In these equations, $z(drug,x)$ represents the z-score (differential expression) of the gene $x$ observed after the drug administration; $z(disease,x)$ represents the z-score of the gene $x$ observed in disease condition. $\|z(drug,x)+z(disease,x)\|$ gives the absolute value of the sum of these differential gene expression values. The main purpose of this special formulation is to increase the value of Adamic-Adar coefficient ($S_x$). When the z-scores of a gene for a drug and a disease would be in a reverse direction (*e.g.*, one is up-regulated

and the other is down-regulated), their sum would be close to 0, which is a desirable effect of drug-treatment in the cell (*Chen et al., 2017*). Thus, if the expression values of common genes in DGN and DAPN are damping each other in terms of RNA-sequencing measurements, this will lead to the ranking of a related gene in higher levels; in other cases it will lower the ranking of the same gene. If the gene of interest is found in only one of the networks (either in DGN or DAPN, *i.e.,* it is not a common gene), the Adamic-Adar coefficient value and the rank of this gene decrease, indicating that this would not be an important gene in terms of the similarity score.

### PageRank

The PageRank algorithm (*Page et al., 1999*) is an adaptation of random walk and it has been widely used for network centrality analysis. The *Page_rank* function in the *igraph* package was used, the performance was tested with different *damping* parameters. Several trials were designed to fix the damping parameter. Finally, it was assigned to 0.75 which was the best performing value. When the specified *Page_rank* function was run with the damping factor value of 0.75, the ranking value obtained for each gene was expressed as $PR(x)$. Equations (5) and 6 show that the integration of differential expression values of genes in the disease and drug-affected networks into the initial $PR(x)$ value, which was applied by the same technique as explained in the Adamic-Adar coefficient.

$$PR_1(x) = \begin{cases} PR(x) * 0.5^{\|z(drug,x)+z(disease,x)\|}, & \text{if } x \in (\text{DGN \& DAPN}) \\ PR(x) * 0.5^{\|z(drug,x)\|}, & \text{if } x \in (\text{DAPN}) \\ PR(x) * 0.5^{\|z(disease,x)\|}, & \text{if } x \in (\text{DGN}) \end{cases} \tag{5}$$

$$PR_2(x) = \begin{cases} PR(x)/e^{\|z(drug,x)+z(disease,x)\|}, & \text{if } x \in (\text{DGN \& DAPN}) \\ PR(x)/e^{\|z(drug,x)\|}, & \text{if } x \in (\text{DAPN}) \\ PR(x)/e^{\|z(disease,x)\|}, & \text{if } x \in (\text{DGN}) \end{cases} \tag{6}$$

### Neighborhood scoring

This metric prioritizes the distribution of differentially expressed genes within the network structure, it considers differential expression value of each gene in terms of fold-change value (*Nitsch et al., 2010*). The neighborhood scoring of a gene $i$ is:

$$X_i = \alpha . x_i + (1 - \alpha) * (\sum_{j \neq i, j = w_{ij} > \epsilon} x_j)/N \tag{7}$$

where $x_i$ is the z-score of the gene $i$; $x_j$ is the z-score of the neighbors of $i$; $N$ indicates the total number of neighbors of $i$; $\alpha$ is a fixed value between 0-1, which indicates a threshold value for interaction between $i$ and its neighbors. $w_{ij}$ denotes the interaction weight between genes $i$ and $j$. The value $\epsilon$ indicates the selected threshold value for the $w_{ij}$ weight. The FIN network was updated by removing edges with link weights less than 0.1. For this reason, the $\epsilon$ threshold value is accepted as 0.1 in here. Different $\alpha$ values were tested and set to 0.7.

## Validation and AUC calculation

Each metric applying different biological hypotheses calculates a score/rank value for each protein in the network; the network can be the drug-affected protein network (DAPN) or the disease gene network (DGN). All proteins in the DAPN were ordered by high to low scores. This sorted list was examined separately for each drug. Proteins were checked whether their rank was above and below a threshold value (*e.g.*, $top_{100}$ proteins) which were a member of the DGN module. The total number of such proteins, which were ranked above the $top_p$ and were a member of the DGN, were recorded as true positive (TP) proteins. True negatives (TN) were the proteins that had a lower rank than $top_p$ and did not contain genes in the DGN. False positive (FP) proteins were listed higher than the $top_p$ and did not contain genes in the DGN. False negatives (FN) were listed lower than the $top_p$ and contained genes in the DGN. For each drug, TP, TN, FP, and FN values were determined in the sorted score list according to a threshold value. These values are then used to complete a single confusion matrix. The confusion matrix refers to a single point in the receiver operating characteristic (ROC) curve when True Positive rate (TPR) and False Positive rate (FPR) were computed. Total of 100 different $top_p$ threshold values (*e.g.*, 1%, 2%, …,100% of total number of genes available in each DAPN) were applied to obtain 100 individual measurements in the ROC curve. Then an area under the ROC curve (AUC) score was computed for each drug. This score was called $AUC_{DAPN}$ when DAPN was used in the metrics and DGN was used as the seed (reference set) for creating the confusion matrix. This method was implemented for the metrics run on DGN, and DAPNs became the seed to compute $AUC_{DGN}$. Finally, individual $AUC_{DAPN}$ and $AUC_{DGN}$ scores were integrated to compute a combined AUC score as given in Eq. (8).

$$Combined.AUC = \sqrt{AUC_{DAPN} * AUC_{DGN}}. \tag{8}$$

For each drug used to treat a cancer cell line, a *Combined.AUC* score was obtained, the *Combined.AUC* scores were sorted from the highest to lowest value for each network metric separately. As a result of the adaptations applied on the Adamic-Adar coefficient and PageRank formulas, the ranking of genes in the AUC calculations should be in a decreasing order. However, in the neighborhood scoring metric, a gene may be neutralized by neighboring genes according to their z-score expression values. For this reason, scores of genes better represent the biological hypothesis when they are ordered from the smallest to largest in the AUC calculations. Unlike other metrics, the ranking of genes as an increasing order became more accurate when calculating $AUC_{DGN}$ and $AUC_{DAPN}$ for the neighborhood scoring metric. The other evaluations metrics (F1.score, precision, recall) were also computed with the same adaptation applied for the *Combined.AUC* score.

## Other DR methods

In order to compare the proposed method with other DR methods, the MNBDR approach (*Chen & Zhou, 2021*), SAveRUNNER (*Fiscon & Paci, 2021*) and OCTAD (*Zeng et al., 2021*) were used. MNBDR identifies dense modules in a protein protein interaction network and selects significant modules based on a high number of cross-talks among the dense modules. Later, the PageRank algorithm chooses the important modules of a disease.

Gene expression data of drug treated samples were integrated into significant modules to calculate a DR score for a drug-disease pair. To make a fair evaluation, the original FIN and drug-treated samples of three cancers were used in this experiment. Therefore, MNBDR was run on the same interaction network (FIN) and gene expression data of the drug-targeted cells used in this study. The second DR approach is a network-based tool named SAveRUNNER. It prioritizes potential drugs that are in close neighborhood of disease genes in the network. It also uses a clustering method to update network similarity. The drug targets used to run SAveRUNNER in the current study were retrieved from DrugBank (*Wishart et al., 2018*). The remaining drugs that did not have target proteins were searched over the STITCH database (*Szklarczyk et al., 2015*). The human interactome used in SAveRUNNER was replaced by FIN. The third DR approach was OCTAD, which proposes drugs to target cancer patient groups based on their gene expression profiles. It does not apply a network-based analysis but it uses the drug-treated cancer cell lines data of the LINCS project (*Subramanian et al., 2017*), similar to the current study. OCTAD was run on its web tool by selecting the colorectal cancer, prostate cancer, and melanoma cohorts of TCGA. In order to select significant results, the threshold for the sRGES score of OCTAD was set to $-0.25$ or smaller values, which was the suggested cutoff in its manual.

## RESULTS AND DISCUSSION

The experimental results of the new DR method are presented in this section to suggest new treatment candidates for three cancer types.

### Colorectal cancer

After applying the optimal dosage on drug-treated colorectal cancer samples, 260 drugs were obtained as repositioning candidates. The combined AUC values of the candidate drugs were computed after computing all network metrics on the DAPNs and the DGN of colorectal cancer. There were eight and 12 drugs listed by Adamic-Adar-1 and Adamic-Adar-2, respectively (Table 1) when the significance threshold was set to 0.6 combined AUC value. Based on the results of Adamic-Adar-2, some of drugs such as "*PHA-793887*", "*gefitinib*" and "*nelarabine*", had better combined AUC values (with an increase of 0.01). When the significance threshold was set to 0.6 combined AUC value, there are one and six drugs reported by PageRank-1 and PageRank-2, respectively (Table 2). Based on these results, PageRank-2 had more significant results than PageRank-1. Table 3 shows the results of neighborhood scoring metric, returned 17 drugs above the 0.6 combined AUC value. The new metric versions (Adamic-Adar-2 and PageRank-2) of metrics have achieved better results than their initial versions (Adamic-Adar-1 and PageRank-1). On the other hand, neighborhood scoring predicted the highest number of drugs above the 0.6 AUC threshold. The other evaluation measures (F1-score, precision, recall) for candidate drugs are given in Table S9 for the three network metrics. When the optimal IC50 dosages of the top predicted drugs were analyzed, the IC50 values were almost equally distributed between five optimal IC50 dosages (0.04, 0.12, 0.37, 1.11, 3.33, and 10 µM) for each metric (Figs. S2A, S2B, S2C).

**Table 1** DR candidates suggested by the Adamic-Adar-1 and Adamic-Adar-2 metrics for colorectal cancer.

| | Adamic-Adar-1 | | Adamic-Adar-2 | |
|---|---|---|---|---|
| Rank | Drug | Combined.AUC | Drug | Combined.AUC |
| 1 | PHA-793887 | 0.63 | PHA-793887 | 0.64 |
| 2 | gefitinib | 0.62 | gefitinib | 0.63 |
| 3 | BX-795 | 0.61 | nelarabine | 0.62 |
| 4 | pterostilbene | 0.61 | pterostilbene | 0.62 |
| 5 | nelarabine | 0.61 | BX-795 | 0.61 |
| 6 | olaparib | 0.61 | Y-39983 | 0.61 |
| 7 | lapatinib | 0.60 | gefitinib | 0.61 |
| 8 | enoxolone | 0.60 | enoxolone | 0.61 |
| 9 | | | alisertib | 0.61 |
| 10 | | | AT-7519 | 0.60 |
| 11 | | | ZM-447439 | 0.60 |
| 12 | | | KIN001-244 | 0.60 |

**Table 2** DR candidates suggested by the Page Rank-1 and Page Rank-2 metrics for colorectal cancer.

| | PageRank-1 | | PageRank-2 | |
|---|---|---|---|---|
| Rank | Drug | Combined.AUC | Drug | Combined.AUC |
| 1 | AT-7519 | 0.60 | AT-7519 | 0.61 |
| 2 | | | dabrafenib | 0.60 |
| 3 | | | cytarabine | 0.60 |
| 4 | | | alectinib | 0.60 |
| 5 | | | dasatinib | 0.60 |
| 6 | | | Y-39983 | 0.60 |

We systematically searched the ClinicalTrials.gov (http://clinicaltrials.gov) website to explore phase trials that run over the predictions of new DR method. There were some clinical phase trials for two drugs *(gefitinib, alisertib)* predicted by the Adamic-Adar-2 metric and three drugs *(dabrafenib, alectinib, dasatinib)* suggested by the Page Rank-2 metric. Seven drugs *(quercetin, sorafenib, tozasertib, taselisib, venetoclax, gemcitabine, pazopanib)* predicted by neighborhood scoring had reported clinical trials on colorectal cancer. Some of these predictions of new DR method also showed promising clinical results (these include: *dabrafenib: NCT04294160, NCT04294160,* (*Al-Taie et al., 2021*); *alectinib: NCT04644315; dasatinib: NCT00920868* (*Strickler et al., 2013*; *Lucchetta & Pellegrini, 2021*; *Al-Taie et al., 2021*; *Scott et al., 2017*); *alisertib:* (*Manfredi et al., 2011*; *Cervantes et al., 2012*; *Stathias et al., 2018*); *pazobanip: NCT00387387* (*Brady et al., 2013*); *gefitinib: NCT00026364* (*Meyerhardt et al., 2007*; *Wolpin et al., 2006*), *NCT00025142* (*Fisher et al., 2008*; *Wang et al., 2019*; *Al-Taie et al., 2021*); *sorafenib: NCT00826540* (*Xie et al., 2020*; *Lucchetta & Pellegrini, 2021*). These compounds have been suggested in several studies and clinical trials as new treatment alternatives for colorectal cancer.

**Table 3    DR candidates suggested by the neighborhood scoring metric for colorectal cancer.**

| | Neighborhood scoring | |
| --- | --- | --- |
| Rank | Drug | Combined.AUC |
| 1 | GSK-1904529a | 0.66 |
| 2 | KIN001-266 | 0.65 |
| 3 | pilaralisib | 0.62 |
| 4 | quercetin | 0.62 |
| 5 | leflunomide | 0.62 |
| 6 | LDN-193189 | 0.62 |
| 7 | sorafenib | 0.61 |
| 8 | tozasertib | 0.61 |
| 9 | ZSTK-474 | 0.61 |
| 10 | amsacrine | 0.61 |
| 11 | taselisib | 0.60 |
| 12 | venetoclax | 0.60 |
| 13 | gemcitabine | 0.60 |
| 14 | apabetalone | 0.60 |
| 15 | quizartinib | 0.60 |
| 16 | pazopanib | 0.60 |
| 17 | SB-590885 | 0.60 |

## Prostate cancer

The DR method was run on 243 different drugs for prostate cancer. When the combined AUC value was set to 0.6, the Adamic-Adar-2 metric recommended twice as many drugs than Adamic-Adar-1 (Table 4). Adamic-Adar-2 estimated 13 drugs, while Adamic-Adar-1 found six drugs. *Tamoxifen* was reported to be the best repurposed candidate with a 0.77 AUC by the Adamic-Adar-1 metric, however, it had an even higher score (0.81 AUC) by the Adamic-Adar-2 metric. There were 10 and 14 drugs reported by PageRank-1 and PageRank-2, respectively (Table 5). The AUC values increased by 0.02 on a drug basis in the PageRank-2 metric. PageRank metrics also reported an FDA-approved treatment (*rucaparib*) for prostate cancer. Therefore, random-walk based metrics were shown to propose already-approved treatments and can indicate the reliability of suggested drugs as new treatment candidates. For prostate cancer results, the new metric versions (Adamic-Adar-2 and PageRank-2) gave more promising results than the initial testing methods. The neighborhood scoring metric estimated 14 drugs to be on the same threshold, which was similar to the number of drugs predicted by PageRank-2. *Docetaxel*, which is one of the FDA approved drugs for prostate cancer, is ranked between the results of neighborhood scoring (Table 6). The other evaluation measures (F1-score, precision, recall) for candidate drugs are given in Table S10 for the three network metrics. The optimal IC50 dosages of the top predicted drugs were almost equally distributed between five optimal IC50 dosages for each metric (Figs. S2D, S2E, S2F).

A search for clinical trials on prostate cancer revealed that five drugs *(tamoxifen, NVP-BEZ235, tretinoin, vorinostat, palbociclib)* predicted by the Adamic-Adar-2 metric

**Table 4** DR candidates suggested by the Adamic-Adar-1 and Adamic-Adar-2 metrics for prostate cancer.

| | Adamic-Adar-1 | | Adamic-Adar-2 | |
|---|---|---|---|---|
| Rank | Drug | Combined.AUC | Drug | Combined.AUC |
| 1 | tamoxifen | 0.77 | tamoxifen | 0.81 |
| 2 | phenformin | 0.71 | phenformin | 0.70 |
| 3 | naftopidil | 0.66 | naftopidil | 0.63 |
| 4 | GSK-1904529A | 0.61 | GSK-1904529A | 0.63 |
| 5 | QL-XII-47 | 0.61 | QL-XII-47 | 0.62 |
| 6 | NVP-TAE226 | 0.60 | NVP-BEZ235 | 0.62 |
| 7 | | | tretinoin | 0.61 |
| 8 | | | vorinostat | 0.61 |
| 9 | | | NVP-TAE226 | 0.61 |
| 10 | | | AZD-7762 | 0.60 |
| 11 | | | ACY-1215 | 0.60 |
| 12 | | | JW-7-24-1 | 0.60 |
| 13 | | | palbociclib | 0.60 |

**Table 5** DR candidates suggested by the Page Rank-1 and Page Rank-2 metrics for prostate cancer (FDA-approved drugs are shown in bold).

| | PageRank-1 | | PageRank-2 | |
|---|---|---|---|---|
| Rank | Drug | Combined AUC | Drug | Combined.AUC |
| 1 | GSK-1904529A | 0.69 | GSK-1904529A | 0.70 |
| 2 | ACY-1215 | 0.65 | ACY-1215 | 0.68 |
| 3 | NVP-BEZ235 | 0.62 | NVP-BEZ235 | 0.64 |
| 4 | GSK-690693 | 0.62 | GSK-690693 | 0.64 |
| 5 | MG-132 | 0.61 | **rucaparib** | 0.63 |
| 6 | AGI-5198 | 0.61 | linsitinib | 0.62 |
| 7 | linsitinib | 0.61 | MG-132 | 0.62 |
| 8 | TAK-715 | 0.60 | AGI-5198 | 0.62 |
| 9 | **rucaparib** | 0.60 | serdemetan | 0.62 |
| 10 | NVP-BEZ235 | 0.60 | TAK-715 | 0.62 |
| 11 | | | NVP-BEZ235 | 0.61 |
| 12 | | | entinostat | 0.60 |
| 13 | | | AZD-6482 | 0.60 |
| 14 | | | voxtalisib | 0.60 |

and three drugs *(NVP-BEZ235, linsitinib, entinostat)* suggested by Page Rank-2 have been reported in different clinical trials. The neighborhood scoring metric reported four drugs *(irinotecan, gemcitabine, etoposide, linsitinib)* that have been observed in clinical trials as well. Several predictions of the proposed model have promising validations in both computational studies and clinical trials as new therapy alternatives for prostate cancer (these include: *etinostat:* (*Turanli & Arga, 2019*); *etoposide:* NCT02861573, NCT03582475 (*Luo et al., 2021*); *gemcitabine:* NCT00014456 (*Dragnev et al., 2010*; *Bibby et al., 2021*);

**Table 6** DR candidates suggested by the neighborhood scoring metric for prostate cancer (FDA-approved drugs are shown in bold).

| | Neighborhood scoring | |
| --- | --- | --- |
| Rank | Drug | Combined.AUC |
| 1 | phenformin | 0.69 |
| 2 | **docetaxel** | 0.65 |
| 3 | QL-XII-47 | 0.64 |
| 4 | 2126458 | 0.64 |
| 5 | NU-7441 | 0.63 |
| 6 | irinotecan | 0.63 |
| 7 | 1904529A | 0.63 |
| 8 | gemcitabine | 0.62 |
| 9 | ciprofloxacin | 0.62 |
| 10 | PHA-793887 | 0.61 |
| 11 | etoposide | 0.61 |
| 12 | UNC-1215 | 0.61 |
| 13 | OSI-027 | 0.60 |
| 14 | linsitinib | 0.60 |

*irinotecan:* (*Wissing et al., 2013*); *palbociclib: NCT04606446, NCT03878524* (*Wang et al., 2019*); *tretinoin: NCT03878524; vorinostat: NCT03878524* (*Turanli & Arga, 2019*).

## Melanoma

The DR method tested 220 different drugs for melanoma. When the significance threshold was set to 0.6, there were 66 and 72 drugs suggested by Adamic-Adar-1 and Adamic-Adar-2, respectively. Due to difficulty of analyzing such a long candidate list, the combined AUC threshold was set to 0.7. Ultimately, 12 and 13 drugs were reported by Adamic-Adar-1 and Adamic-Adar-2, respectively (Table 7). Although the two metrics suggested an almost equal number of candidates, Adamic-Adar-2 had slightly higher AUC values. Additionally, Adamic-Adar-2 predicted one FDA-approved drug (*trametinib*) for melanoma treatment with a 0.72 AUC value. When the combined AUC value was set to 0.7, it resulted in five and seven drugs proposed by PageRank-1 and PageRank-2, respectively (Table 8). The PageRank-2 metric achieved slightly higher AUC values than the PageRank-1, which was similar to results of other cancers. Contrary to other cancer types, neighborhood scoring metric could not report as many drugs as other metrics for melanoma (Table 9); ultimately, it only reported four drugs. The optimal IC50 dosages of the top predictions were almost equally distributed between five optimal IC50 dosages for each metric (Figs. S2G, S2H, S2I). The other evaluation measures (F1-score, precision, recall) for candidate drugs are given in Table S11 for three network metrics.

There are several phase trials for three drugs *(dasatinib, navitoclax, vorinostat)* listed by Adamic-Adar-2 (Table 7), four drugs *(vorinostat, navitoclax, dinaciclib, MK-1775)* listed by PageRank-2 (Table 8) and all the drugs *(paclitaxel, imexon, sulforaphane, vorinostat)* predicted by neighborhood scoring (Table 9) for the treatment of melanoma. Some of these predictions suggested by the new DR method have been used in several studies and

**Table 7  DR candidates suggested by the Adamic-Adar-1 and Adamic-Adar-2 metrics for melanoma (FDA approved drugs shown in bold).**

| | Adamic-Adar-1 | | Adamic-Adar-2 | |
|---|---|---|---|---|
| Rank | Drug | Combined.AUC | Drug | Combined.AUC |
| 1 | dasatinib | 0.83 | dasatinib | 0.86 |
| 2 | sulforaphane | 0.81 | sulforaphane | 0.77 |
| 3 | avagacestat | 0.80 | avagacestat | 0.77 |
| 4 | tivozanib | 0.74 | tivozanib | 0.75 |
| 5 | ZM-447439 | 0.74 | pevonedistat | 0.75 |
| 6 | elesclomol | 0.73 | ZM-447439 | 0.74 |
| 7 | pevonedistat | 0.73 | betulinic-acid | 0.74 |
| 8 | betulinic-acid | 0.72 | elesclomol | 0.72 |
| 9 | vorinostat | 0.71 | navitoclax | 0.72 |
| 10 | OSI-930 | 0.70 | **trametinib** | 0.72 |
| 11 | navitoclax | 0.70 | vorinostat | 0.71 |
| 12 | cytarabine | 0.70 | navitoclax | 0.70 |
| 13 | cediranib | 0.70 | | |

**Table 8  DR candidates suggested by the Page Rank-1 and Page Rank-2 metrics for melanoma.**

| | PageRank-1 | | PageRank-2 | |
|---|---|---|---|---|
| Rank | Drug | Combined.AUC | Drug | Combined.AUC |
| 1 | vorinostat | 0.75 | vorinostat | 0.75 |
| 2 | cytarabine | 0.73 | cytarabine | 0.74 |
| 3 | MK-1775 | 0.71 | MK-1775 | 0.73 |
| 4 | sulforaphane | 0.70 | navitoclax | 0.71 |
| 5 | navitoclax | 0.70 | tivozanib | 0.70 |
| 6 | | | amonafide | 0.70 |
| 7 | | | dinaciclib | 0.70 |

**Table 9  DR candidates suggested by the neighborhood scoring metric for melanoma.**

| | Neighborhood scoring | |
|---|---|---|
| Rank | Drug | Combined.AUC |
| 1 | paclitaxel | 0.78 |
| 2 | imexon | 0.75 |
| 3 | sulforaphane | 0.74 |
| 4 | vorinostat | 0.70 |

clinical trials in the search for new treatment alternatives for melanoma(these include: *vorinostat: NCT00121225 (Haas et al., 2014; Choi et al., 2022; Wang et al., 2018; Nihal, Roelke & Wood, 2010); MK-1775: (Margue et al., 2019); imexon: NCT00327600 (Weber et al., 2010); paclitaxel: NCT01107665 (Fruehauf et al., 2018).*

## Comparison with other DR methods

The proposed method was compared with three DR models: MNBDR (*Chen & Zhou, 2021*), SAveRUNNER (*Fiscon & Paci, 2021*) and OCTAD (*Zeng et al., 2021*).

### MNBDR

The results of this study were first compared with MNBDR which also used protein-protein interactions and gene expression profiles (*Chen & Zhou, 2021*). We used the top 10% predictions of both DR approaches to make a fair evaluation.

The top 10% of the predictions resulted in 26 candidate drugs for the treatment of colon cancer. PageRank-2 identified four common drugs (*dasatinib, dinaciclib, PF-562271, BMS-345541*) while Adamic-Adar-2 (*PF-562271, BMS-345541, dinaciclib*) and neighborhood scoring (*Ro-4987655, dasatinib, alpelisib*) showed three mutual drugs using the MNBDR method (Table S1).

A total of 24 drugs in the ranked lists for prostate cancer were compared. PageRank-2 had five mutual drugs (*NVP-BEZ235, MG-132, voxtalisib, YM-155, mitoxantrone*) with MNBDR, while Adamic-Adar-2 had four common drugs (*NVP-BEZ235, PHA-793887, mitoxantrone, JNK-9L*). Although Adamic-Adar-2 and PageRank-2 identified two FDA-approved drugs (*rucaparib, mitoxantrone*) for prostate cancer, MNBDR suggested only one drug (*mitoxantrone*) in the top-ranked predictions. Neighborhood scoring predicted one FDA-approved drug (*docetaxel*) and two mutual drugs (*irinotecan, PHA-793887*) using MNBDR (Table S2).

PageRank-2 predicted six mutual drugs (*vorinostat, amonafide, dinaciclib, alisertib, etoposide, SN-38*) with MNBDR in the top-ranked 22 drugs in melanoma. Neighborhood scoring and Adamic-Adar-2 had five mutual drugs (*paclitaxel, vorinostat, PF-562271, SN-38, podophyllotoxin*) and single mutual drug (*vorinostat*) using MNBDR, respectively (Table S3).

### SAveRUNNER

SAveRUNNER was the second method used to compare the results of this study (*Fiscon & Paci, 2021*). This method uses human interactome and disease genes.

For colorectal cancer results, the predictions of this method returned 32 candidate drugs. Out of these predictions, four of them were mutual drugs between two studies. Adamic-Adar-2 identified three common drugs (*enoxolone, gefitinib, sorafenib*); one common drug was listed by PageRank-2 (*dabrafenib*) and neighborhood scoring (*sorafenib*) (Table S4).

A total of 28 drugs were predicted by SAveRUNNER for prostate cancer. Five drugs were mutually predicted by both methods. Adamic-Adar-2 identified two common drugs (*naftopidil, tamoxifen*) and neighborhood scoring listed three common drugs (*ciprofloxacin, etoposide, irinotecan*). PageRank-2 did not identify any common drug with SAveRUNNER (Table S5).

The repositioned drugs for melanoma were very limited, only one drug (*avl-292*) was listed by SAveRUNNER (Table S6). The fewer number of disease genes covered in melanoma may have resulted in such a short list prediction. There is no drug to treat melanoma that was mutually predicted by the current method and SAveRUNNER.

### OCTAD

The third tool used to compare the results of this study was OCTAD (*Zeng et al., 2021*), which retrieves repositioned drugs based on a comparison of expression profiles of disease genes and drugs; it does not integrate any network data in computations.

This method retrieved 214 candidate drugs with significant sRGES scores ($\leq -0.25$) for colorectal cancer. Six of these were mutual drugs between two studies. Neighborhood scoring identified three common drugs (*amsacrine, gemcitabine, ZSTK-474*); two common drugs (*dabrafenib, dasatinib*) were listed by PageRank-2 and one mutual drug (*PHA-793887*) was returned by Adamic-Adar-2 (Table S7).

A total of 471 repositioned drugs were returned by OCTAD with significant scores for prostate cancer. Fourteen drugs were mutual between two studies. Adamic-Adar-2 listed six common drugs (*mitoxantrone, NVP-BEZ235, palbociclib, PHA-793887, tamoxifen, vorinostat*); seven common drugs (*entinostat, MG-132, mitoxantrone, NVP-BEZ235, palbociclib, serdemetan, YM-155*) were returned by PageRank-2 and five mutual drugs (*etoposide, gemcitabine, irinotecan, OSI-027, PHA-793887*) were listed by neighborhood scoring (Table S8).

There were 636 candidate drugs with significant scores for melanoma. Eight of these were mutual drugs between two studies. PageRank-2 identified six common drugs (*amonafide, cytarabine, etoposide, pevonedistat, SN-38, vorinostat*); four common drugs (*cytarabine, elesclomol, pevonedistat, vorinostat*) were listed by Adamic-Adar-2, and four common drugs (*cytarabine, PF-562271, SN-38, vorinostat*) were returned by neighborhood scoring (Table S12).

These results revealed that each metric proposed in this study may list different drugs as potential candidates. When four DR methods were considered, several drugs were mutually predicted at least by three methods (Table 10). *Dabrafenib* and *dasatinib* were the mutual drugs suggested for colorectal cancer. *Etoposide, irinotecan, NVP-BEZ235, MG-132, mitoxantrone, PHA-793887, tamoxifen*, and *YM-155* were other common drugs proposed for prostate cancer. *Amonafide, etoposide, PF-562271, SN-38* and *vorinostat* were mutual drugs suggested for melanoma. Six drugs (*dinaciclib, etoposide, gemcitabine, PHA-793887, PF-562271, vorinostat*) were concurrently proposed as candidate treatments of two cancers. Additionally, almost half of these drugs were also considered in clinical trials for colorectal cancer, prostate cancer, and melanoma, as was shown in previous sections. Hence, these repositioned candidates are quite promising, further experiments should be conducted in a laboratory environment.

## CONCLUSION

CDR is a complex procedure and it should consider the chemical and metabolic effects of drugs and measurements of diseases at transcriptome level. CDR cannot make a final treatment decision, however it can suggest the prospective drug-disease combinations that have higher potentials for treatment. Then, these predictions should be evaluated by wet laboratory experiments in cellular levels and animal models.

In this study, drug-treated transcriptome data from the LINCS project were used for three cancer types (colorectal, prostate, melanoma) individually. RNA-sequencing data for

**Table 10  Mutually suggested candidates by four DR methods (drugs predicted by three or more methods are shown in bold).**

| Cancer type | FDA-Approved | Drug name | Current study | MNBDR | SAveRUNNER | OCTAD |
|---|---|---|---|---|---|---|
| Colon Cancer | | alpelisib | NS | X | | |
| Colon Cancer | | amsacrine | NS | | | X |
| Colon Cancer | | BMS-345541 | AA, PR | X | | |
| Colon Cancer | | **dabrafenib** | PR | | X | X |
| Colon Cancer | | **dasatinib** | PR, NS | X | | X |
| Colon Cancer | | dinaciclib | AA, PR | X | | |
| Colon Cancer | | enoxolone | AA | | X | |
| Colon Cancer | | gefitinib | AA | | X | |
| Colon Cancer | | gemcitabine | NS | | | X |
| Colon Cancer | | PF-562271 | AA, PR | X | | |
| Colon Cancer | | PHA-793887 | AA | | | X |
| Colon Cancer | | RO-4987655 | NS | X | | |
| Colon Cancer | | sorafenib | NS | | X | |
| Colon Cancer | | ZSTK-474 | NS | | | X |
| Prostate Cancer | | ciprofloxacin | NS | | X | |
| Prostate Cancer | FDA-Approved | docetaxel | NS | | | |
| Prostate Cancer | | entinostat | PR | | | X |
| Prostate Cancer | | **etoposide** | NS | | X | X |
| Prostate Cancer | | gemcitabine | NS | | | X |
| Prostate Cancer | | **irinotecan** | NS | X | X | X |
| Prostate Cancer | | JNK-9L | AA | X | | |
| Prostate Cancer | | **MG-132** | PR | X | | X |
| Prostate Cancer | FDA-Approved | **mitoxantrone** | AA, PR | X | | X |
| Prostate Cancer | | naftopidil | AA | | X | |
| Prostate Cancer | | **NVP-BEZ235** | AA, PR | X | | X |
| Prostate Cancer | | OSI-027 | NS | | | X |
| Prostate Cancer | | palbociclib | AA | | | X |
| Prostate Cancer | | **PHA-793887** | AA, NS | X | | X |
| Prostate Cancer | | serdemetan | PR | | | X |
| Prostate Cancer | | **tamoxifen** | AA | | X | X |
| Prostate Cancer | | vorinostat | AA | | | X |
| Prostate Cancer | | voxtalisib | PR | X | | |
| Prostate Cancer | | **YM-155** | PR | X | | X |
| Melanoma | | **amonafide** | PR | X | | X |
| Melanoma | | alisertib | PR | X | | |
| Melanoma | | cytarabine | AA, PR, NS | | | X |
| Melanoma | | dinaciclib | PR | X | | |
| Melanoma | | elesclomol | AA | | | X |
| Melanoma | | **etoposide** | PR | X | | X |
| Melanoma | | paclitaxel | NS | X | | |
| Melanoma | | pevonedistat | AA, PR | | | X |

**Table 10** (*continued*)

| Cancer type | FDA-Approved | Drug name | Current study | MNBDR | SAveRUNNER | OCTAD |
|---|---|---|---|---|---|---|
| Melanoma | | **PF-562271** | NS | X | | X |
| Melanoma | | podophyllotoxin | NS | X | | |
| Melanoma | | **SN-38** | PR, NS | X | | X |
| Melanoma | FDA-Approved | trametinib | AA | X | | |
| Melanoma | | **vorinostat** | AA, PR, NS | X | | X |

**Notes.**

AA, Adamic-Adar; NS, Neighborhood Scoring; PR, PageRank.

these cancer types were obtained from the TCGA project. To the best of our knowledge, the construction of individual drug-affected network modules is a novel contribution for CDR. The application of network neighborhood metrics to compute a similarity between networks is another unique adaptation in this domain. The specific z-score adaptations integrated into these metrics made significant improvements in predictions of these metrics. All metrics predicted several drugs above the determined AUC threshold. Some of the suggested drugs were approved by either clinical phase trials or other *in-vivo*/animal studies. Based on these contributions, the proposed CDR method has yielded promising results for different cancer types. As a summary, this novel CDR method should be considered alongside more traditional computational treatment recommendation approaches.

This study demonstrates that the proposed method can be applied for different size of data sets. In the future, a new functional interaction network may be adapted for different purposes. Different weight attributes may be integrated on metrics rather than gene expression values or new network analysis metrics might be easily tested on this process. The proposed method is open for enrichment in terms of different adaptations.

### Funding

This study was supported by The Scientific and Technological Research Council of Turkey (TUBITAK) with the project number 318S276. The funders had no role in study design, data collection and analysis, decision to publish, or preparation of the manuscript.

### Grant Disclosures

The following grant information was disclosed by the authors:
The Scientific and Technological Research Council of Turkey (TUBITAK): 318S276.

### Competing Interests

The authors declare there are no competing interests.

### Author Contributions

- Ali Cüvitoğlu conceived and designed the experiments, performed the experiments, analyzed the data, prepared figures and/or tables, authored or reviewed drafts of the article, and approved the final draft.

- Zerrin Isik conceived and designed the experiments, analyzed the data, prepared figures and/or tables, authored or reviewed drafts of the article, and approved the final draft.

## Data Availability

The drug experiment data is available in NCBI GEO: GSE70138.

The cancer patient data is available in GDC portal under the specified repositories: https://portal.gdc.cancer.gov/projects/TCGA-COAD. https://portal.gdc.cancer.gov/projects/TCGA-PRAD. https://portal.gdc.cancer.gov/projects/TCGA-SKCM.

## Supplemental Information

Supplemental information for this article can be found online at http://dx.doi.org/10.7717/peerj.15624#supplemental-information.

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
