# Peer review of "Network neighborhood operates as a drug repositioning method for cancer treatment"

_PeerJ, doi:10.7717/peerj.15624_

## Round 0.1 · original submission · Major Revisions

Please exhaustively address all the points raised by the referees.

Reviewer 1 ·

Basic reporting

1- There is no comparison with other state of the art algorithms/tools.
Some important similar methods
https://doi.org/10.1186/s12967-020-02541-3

https://doi.org/10.1002/psp4.12670

https://doi.org/10.1093/bib/bbab319

Experimental design

pass

Validity of the findings

2- AUC is a very important performance measures for evaluating the repurposed drugs. However, other perf. mea. Like ACC, Spec, SN, … is also very useful to have a clear and unbiased picture of a model's performance.

Additional comments

In the present study, Cuvitoglu et. al. used Network neighborhood for drug repositioning. They also applied the method on melanoma, colorectal and prostate cancers in which Several candidate drugs were predicted by applying 0.6 or higher AUC values. The problem if of outmost important. Below are my comments:

1- There is no comparison with other state of the art algorithms/tools.
Some important similar methods
https://doi.org/10.1186/s12967-020-02541-3

https://doi.org/10.1002/psp4.12670

https://doi.org/10.1093/bib/bbab319

2- AUC is a very important performance measures for evaluating the repurposed drugs. However, other perf. mea. Like ACC, Spec, SN, … is also very useful to have a clear and unbiased picture of a model's performance.
3- Figure 1 doesn't have clear and informative caption. For example authors can use color coded nodes instead of using arrows to pint to some nodes.
The manuscript suffers from informative visualizations.

Reviewer 2 ·

Basic reporting

English is not clear and professional and need to be revised throughly.

Experimental design

1- The authors should provide more information about their hypothesis. Why should DGN and DPAN be so similar while we know that a reasonable treatment should increase the expression of disease genes that are up-regulated and decrease the expression of disease genes that are down-regulated.

2- Authors need to clarify method section.
For example: “Our DR model is based on several network structures” it seems that they just use FIN network!

Validity of the findings

The method has been tested on melanoma, colorectal cancer, and prostate cancer. What is the correlation between IC50/EC50 and drug combined.auc values?

There is no comprehensive comparison with the previous methods.

Additional comments

In table 1, there are two different combined.auc values for gefitinib (0.63 and 0.61).

·

Basic reporting

1) The English language is properly used so that an international audience can follow the text. Some typos that need to be corrected are as following:
- In lines 228 and 229, 232, 237 space is missing before AUC.
- In line 83, Instead of “1 METHOD” it should be just “METHOD”
In line 222, instead of “2 RESULTS AND DISCUSSION” it should be just “RESULTS AND DISCUSSION”.

2) This article includes some introduction and background on the significance of computational drug repurposing studies, especially network based studies, and why it is needed to repurpose existing drugs. However, computational drug repositioning is also widely used for COVID-19. This needs to be added into the Introduction section and related papers need to be cited. There are several network-based and signature-based (transcriptome profiling, expression based) Computational Drug Repositioning studies conducted for COVID-19. Especially these need to be referred and how this study differs from those studies needs to be emphasized.

3) In terms of related work, the authors need to give more detail of the other cited studies. For example, in line 49 they state that “They reached a 0.92 AUC in their experiments.” Which dataset is used here, related to which disease, which cross validation technique is used? LOOCV, n-fold, etc? For other related work, these details should be better provided.

4) The structure of the article conforms to the standard sections. Figure 1 and the tables of the manuscript are relevant in general and their qualities are OK.

5) The manuscript is self-contained.

Experimental design

The article is scientifically and methodologically sound. The submission clearly defines the research question, which is relevant and meaningful. The proposed method was tested on melanoma, colorectal and prostate cancer datasets.

Validity of the findings

The authors state that “Several candidate drugs were predicted by applying 0.6 or higher AUC values.” AUC value threshold of 0.6 may be low. Do other competitor methods also get low AUC values for those datasets?

Some of the predictions were approved by clinical phase trials or other in-vivo studies found in literature.

Additional comments

The use of "our", "we" throughout the manuscript makes the writing quite informal and I would recommend that this be changed throughout. The manuscript is the presentation of a scientific investigation. It is not owned by, nor belongs to, any set of authors as the same study can be proposed and implemented by others. Therefore, it is more appropriate to refer to "The study", or "the results" and what explicitly occurred during the investigation, rather than "Our study" "Our dataset", "we applied" etc..

---

## Round 0.2 · Minor Revisions

Please address all issues raised.

Reviewer 2 ·

Basic reporting

All concerns have been responded.

Experimental design

pass

Validity of the findings

pass

Reviewer 4 ·

Basic reporting

no comment

Experimental design

If possible it could be useful to descrive in a more specific way what the authors define as DEGs inside chapter "Disease-Causing Genes", in particular the groups used to make the comparison and get the fold change (group 1 vs group 2).

Validity of the findings

no comment

Additional comments

Maybe one simple table (in the main text, inside the results and discussions or Conclusions paragraphs) with the name of drugs predicted by mutliple methods in each dataset (when applicable/useful) could help to summarize the results.

EG

Dataset Clinical use Drug Method 1 Method 2 Method3

Colorectal cancer Y drug_1 X X
N drug_2 X X

Melanoma Y drug_3 X X X
........

---

## Round 0.3 · Minor Revisions

The author have adequately addressed all points raised by the referees. English language needs editing.

---

## Round 0.4 · accepted · Accept

The authors have adequately addressed all issues raised and extensively revised the manuscript which is now acceptable for publication.